# Ultraviolet-C light at 222 nm has a high disinfecting spectrum in environments contaminated by infectious pathogens, including SARS-CoV-2

**Byeong-Min Song[1]⊙, Gun-Hee Lee[1]⊙, Hee-Jeong Han[1], Ju-Hee Yang[1], Eun-Gyeong Lee[1], Hyunji Gu[1], Ha-Kyeong Park[1], Kyunga Ryu[2], Jinwoo Kim[2], Sang-Min Kang❶[1]\*, Dongseob Tark[1]\***

1 Laboratory for Infectious Disease Prevention, Korea Zoonosis Research Institute, Jeonbuk National University, Iksan, Republic of Korea, 2 Biodech Inc., Kyonggi University, Gwanggyosan-ro, Yeongtong-gu, Suwon-si, Gyeonggi-do, Republic of Korea

⊙ These authors contributed equally to this work.
\* sangminkang@jbnu.ac.kr (SMK); tarkds@jbnu.ac.kr (DT)

**Data Availability Statement:** we provide the original uncropped and unadjusted image and blot results reported in a "original figure raw image" file.

## Abstract

Ultraviolet light (UV) acts as a powerful disinfectant and can prevent contamination of personal hygiene from various contaminated environments. The 222-nm wavelength of UV-C has a highly effective sterilization activity and is safer than 275-nm UV-C. We investigated the irradiation efficacy of 222-nm UV-C against contaminating bacteria and viruses in liquid and fabric environments. We conducted colony-forming unit assays to determine the number of viable cells and a 50% tissue culture infectious dose assay to evaluate the virus titration. A minimum dose of 27 mJ/cm$^2$ of 222-nm UV-C was required for >95% germicidal activity for gram-negative and -positive bacteria. A 25.1 mJ/cm$^2$ dose could ensure >95% virucidal activity against low-pathogenic avian influenza virus and severe acute respiratory syndrome coronavirus (SARS-CoV-2). In addition, this energy dose of 222-nm UV-C effectively inactivated SARS-CoV-2 variants, Delta and Omicron. These results provide valuable information on the disinfection efficiency of 222-nm UV-C in bacterial and virus-contaminated environments and can also develop into a powerful tool for individual hygiene.

## Introduction

Ultraviolet (UV) irradiation for commercial or medical use has been applied to prevent the transmission of infectious pathogens from contaminated food, water, surface, and air [1, 2]. UV devices have been developed according to their wavelength and application in community health. UV light is classified into UV-A (320–400 nm), UV-B (280–320 nm), and UV-C (200–280 nm) according to wavelength, and each UV light has different properties on sterilization efficacy and skin damage based on the emitted irradiation energy [3, 4]. Short-wavelength UV-C is recognized as a germicidal light and can be used to prevent localized infections for environmentally friendly cleaning due to not requiring any chemicals [5]. UV-C is effective in disinfecting areas contaminated with antibiotic-resistant pathogens [6]. The 254- and 275-nm

**Funding:** This research was supported by the grant of the Korea Health Industry Development Institute (KHIDI, HI22C1637) and Basic Science Research Program through the National Research Foundation of Korea (NRF) funded by the Ministry of Education (2021R1C1C2005307 and 2017R1A6A1A03015876). Sang-Min Kang received a research fund, HI22C1637, from KHIDI and contributed to conceptualizing research, data curation, methodology, supervision, manuscript review, and editing. Gun-Hee Lee received a research fund, 2021R1C1C2005307, from NRF and was involved in Formal analysis, methodology, validation, and manuscript writing. Dongseob Tark received a research fund, 2017R1A6A1A03015876, from NRF and was involved in conceptualization, supervision, validation, and manuscript review.

**Competing interests:** The authors have declared that no competing interests exist.

UV-C are commercial germicidal wavelengths for disinfection. However, these can damage organic matter and the skin or eyes of humans and animals [7]. Consequently, 254- and 275-nm UV-C have only been used in manless environments and are recommended for covering an exposed area. Furthermore, the 222-nm UV-C termed far-UV-C (207–222 nm) demonstrates a similar inactivation effect against infectious pathogens to that of 254-nm UV-C and is less harmful to the skin and eyes in experimental animals [8–10]. This characteristic is caused by the different penetration following the UV-C wavelength, although this is similar to the absorption rate [11]. We chose the 222-nm UV-C to develop a small, portable sterilizer and a human gateway sterilizer that can be directly exposed to humans. The 222-nm wavelength was considered the safest and most effective in the UV-C irradiated human from a close distance. Recently, several studies reported that the aerosolized H1N1 influenza virus and human coronaviruses, including severe acute respiratory syndrome coronavirus 2 (SARS-CoV-2), were inactivated by 222-nm UV-C [12–14]. Both types of the virus produced a pandemic with fast transmission. Among the transmission routes when it spreads to liquid, such as coughing or sneezing, cotton fabrics, such as clothes, are contaminated. Here, we investigate the germicidal effect of 222-nm UV-C against gram-negative and -positive bacteria. Furthermore, we determined the virucidal efficacy of 222-nm UV-C against low-pathogenic avian influenza virus (LPAIV) and SARS-CoV-2 depending on exposure time, distance, and contaminated environments compared with that of 275-nm UV-C. UV-C irradiation at 222-nm effectively disinfects liquid and material contamination by key infectious pathogens, including SARS-CoV-2.

## Materials and methods

### Cell lines

African green monkey kidney (Vero E6) cells and Madin–Darby Canine kidney (MDCK) cells were cultured in Dulbecco's modified Eagle minimum essential medium (DMEM, Gibco, USA) supplemented with 10% fetal bovine serum (FBS, Gibco, USA) and 10 U/mL penicillin/streptomycin (Gibco, USA) at 37°C in a 5% $CO_2$ humidified atmosphere.

### Bacteria and viruses

*S. typhimurium* (KVCC-BA0400600) and *S. aureus* (KVCC-BA200185) were provided by the Animal and Plant Quarantine Agency (APQA, South Korea). Bacteria were cultured aerobically at 37°C to an optical density at 600 nm ($OD_{600}$) of 0.4–0.5 from single colony cultures in Luria–Bertani (LB) culture medium (BD Difco, USA). The number of colony-forming units was measured via serial dilution and viable colony counting. LPAIV H9N2 strain A/chicken/Korea/MS96/1996 (KVCC-VR1100013) was provided by the APQA. LPAIV was propagated through the allantoic cavity route in specific pathogen-free (SPF) embryonated chicken eggs. Inoculated SPF eggs were incubated for 4 days in a 37°C egg incubator. The allantoic fluid containing virus was harvested using a sterile syringe. SARS-CoV-2 Wuhan (BetaCoV/Korea/KCDC03/2020; NCCP 43326), Delta mutant (hCoV-19/Korea/KDCA210812/2021; NCCP 43390), and Omicron (hCoV-19/Korea/KDCA447321/2021; NCCP 43408) stains were provided by the Korea Disease Control and Prevention Agency (KDCA). SARS-CoV-2 was propagated in Vero E6 cells with DMEM supplemented with 2% FBS. The virus was harvested via the freezing-thawing method.

### UV-C irradiation

UV-C equipment consisted of an unfiltered quart tube lamp with an irradiance of 4,360 mW/$cm^2$ at a 10 mm/sec distance from the lamp. The 222-nm wavelength of UV-C was produced

by excimer light technology. The UV-C equipment manufactured for the experiment was provided by the Biodech company in South Korea.

## Biocidal activity test after UV-C irradiation

Bacteria were inoculated as single colonies in LB broth and incubated overnight in a shaking incubator at 37°C. Grown bacteria were seeded in fresh LB broth and incubated at 37°C until the $OD_{600}$ reached 0.5. Bacteria were serially diluted $10^{-1}$ to $10^{-6}$ with LB broth and spread on an LB agar plate. LB agar plates were immediately irradiated with UV-C at differential dosages according to distance. After irradiation, the LB agar plates were incubated overnight at 37°C. The number of colonies was counted and calculated as a $\log_{10}$ CFU/ml.

## Antiviral activity test of UV-C in liquid and fabric conditions

Antiviral test in liquid conditions: LPAIV and SARS-CoV-2 (both at $1 \times 10^6$ $TCID_{50}$/mL) were placed on a sterile culture dish and then irradiated with UV-C at differential dosages according to distance. After irradiation, the viruses were serially diluted with TPCK-DMEM and serum-free DMEM and inoculated into MDCK or Vero E6 cells, respectively. The supernatant containing the virus was replaced with a fresh maintained medium at 1 h post-inoculation. MDCK cells were inoculated for 5 days, and the titration of LPAIV was measured using the hemagglutination (HA) assay. Cytopathic effects (CPEs) were observed in Vero E6 cells infected with SARS-CoV-2 for 3 or 4 days. The titration of SARS-CoV-2 was calculated using the tissue cultured infectious dose 50 ($TCID_{50}$) method as previously described [15]. The antiviral test in fabric conditions: Sterile gauze ($2 \times 2$ cm) was placed in a Petri dish and then LPAIV or SARS-CoV-2 (both at $1 \times 10^7$ $TCID_{50}$/mL) were applied to individual pieces of gauze. Subsequently, the wet gauze was dried at 25°C for complete absorption (15 minute at class II Bio Safety Cabinet). The gauze containing the virus was irradiated with UV-C at differential dosages according to distance. After irradiation, the gauze was transferred to a 5-mL tube containing DMEM without fetal bovine serum and incubated for 1 h at room temperature to elute the virus from the gauze. The virus was serially diluted to inoculate the LPAVI into MDCK and SARS-CoV-2 into Vero E6 cells. Virus titration was calculated via the HA assay or by observing virus-induced cytopathic effects.

## Immunofluorescence assay

Cells were fixed with a fixation buffer containing acetone and methanol in a 1:1 ratio. After 10 min, the cells were incubated with a blocking buffer containing 5% bovine serum albumin. Cells were labeled in blocking solution with rabbit polyclonal antibodies against the spike protein of SARS-CoV-2 (Sino Biologicals US Inc.). Cells were washed with PBS and then labeled with Alexa fluor 488-conjugated anti-rabbit IgG (Cell Signaling Technology, USA) in PBS. Nuclei were stained with 4', 6-diamidino-2-phenylindole (Cell Signaling Technology). Immunofluorescence signals were observed using a CELENA® S digital imaging system (Logos Biosystems, South Korea). The fluorescence intensity quantification was analyzed using HALO image analysis software (HALO v34.2986.185).

## RNA isolation and real-time PCR

Total cellular RNA was extracted from cells infected with SARS-CoV-2 using TRIzol Reagent (Invitrogen, USA). cDNA was synthesized using the CellScript™ cDNA master mix (Cell Signaling Technology, USA). The NSP2 and N genes of SARS-CoV-2 were amplified from the synthesized cDNA using IQ SYBR green Supermix (Bio-Rad, South Korea). The described sets of primers were

used: NSP2 forward primer, 5'-GTA TTG TTA TAG CGG CCT TCT G-3'; NSP2 reverse primer, 5'-ATG CAT TTG CAT CAG AGG CTG C-3'; N gene forward primer, 5'-TAA TCA GAC AAG GAA CTG ATT A-3'; N gene reverse primer, 5'-CGA AGG TGT GAC TTC CAT G-3'. Thermal cycling was performed as follows: initial denaturation at 95°C for 3 min, 40 cycles of denaturation at 95°C for 10 sec, and annealing/extension at 55°C for 30 sec.

## Results

### Germicidal effect of 222-nm UV-C light against gram-negative and -positive bacteria

UV-C equipment manufactured for emittance of UV light at 222 and 275 nm was received from Biodech in South Korea. The energy dosage of both UV-C equipment's over a distance and exposure time is described in Table 1. The germicidal efficacy test of both UV-C wavelengths against two different bacterial species was performed. Briefly, *Salmonella typhimurium* (gram negative) and *Staphylococcus aureus* (gram positive) were cultured in suspension and then spread on an LB agar plate without antibiotics. Subsequently, the bacterial lawn was irradiated with either 222- or 275-nm UV-C at various irradiation doses over different distances. The number of viable bacteria was then counted by colony formation. We then determined that the optimum distance for 222-nm UV-C efficient germicidal activity. The disinfecting activity of UV-C is affected by irradiation time and distance from the lamp. [16, 17]. At 200-mm distance, irradiation for 5 s (85 mJ/cm$^2$) of 275-nm UV-C reduced the bacterial number by 97% for *S. typhimurium* and 98.2% for *S. aureus*. Furthermore, both species were undetectable following irradiation for 20 s (340 mJ/cm$^2$) with 275-nm UV-C (Fig 1A). Irradiation for 60 s (21.12 mJ/cm$^2$) of 222-nm UV-C reduced the bacterial number by 82.27% for *S. typhimurium*; whereas the number of *S. aureus* was reduced 99.5% when irradiated for 30 sec (10.56 mJ/cm$^2$), and was undetectable when irradiated for 60 sec (21.12 mJ/cm$^2$) with 222-nm UV-C (Fig 1A and S1 Table). At 100-mm distance, both bacteria species were undetectable following 275-nm UV-C irradiation for 5 s (520 mJ/cm$^2$). By contrast, 20 s (18 mJ/cm$^2$) of 222-nm UV-C irradiation was required to reduce bacterial numbers to undetectable levels with *S. aureus* and 60 sec (54 mJ/cm$^2$) were required for the same effect on *S. typhimurium* numbers (Fig 1B and S1 Table). Although the germicidal efficiency of 222-nm UV-C was lower than 275-nm UV-C at the same exposure time, it was safer and had a potent germicidal effect that was dosage-dependent in both bacterial species. In addition, we found that *S. aureus* was more sensitive than *S. typhimurium* to irradiation of 222-nm UV-C at the same distance and time.

### Virucidal activity of 222-nm UV-C on LPAIV-contaminated environments

Several studies have been conducted since the development of UV-C (including at 222 nm) that reported the antiviral potential against various viruses [18–20]. Next, we evaluated the virucidal activity of 222-nm UV-C using two different contaminated conditions with LPAIV

**Table 1. The energy dosage of UV-C equipment over a distance and exposure times.**

| Equipment (UV-C) | Distance (mm) | Dosage/sec (mJ/cm$^2$) | Dosage / Exposure time (second) | | | | |
|---|---|---|---|---|---|---|---|
| | | | *5* | *10* | *20* | *30* | *60* |
| 222 nm | 50 | 2.510 | 12.55 | 25.1 | 50.2 | 75.3 | 150.6 |
| | 100 | 0.900 | 4.5 | 9 | 18 | 27 | 54 |
| | 200 | 0.352 | 1.76 | 3.52 | 7.04 | 10.56 | 21.12 |
| 275 nm | 50 | 275 | 1375 | 2750 | 5500 | 8250 | 16500 |
| | 100 | 104 | 520 | 1040 | 2080 | 3120 | 6240 |
| | 200 | 17 | 85 | 170 | 340 | 510 | 1020 |

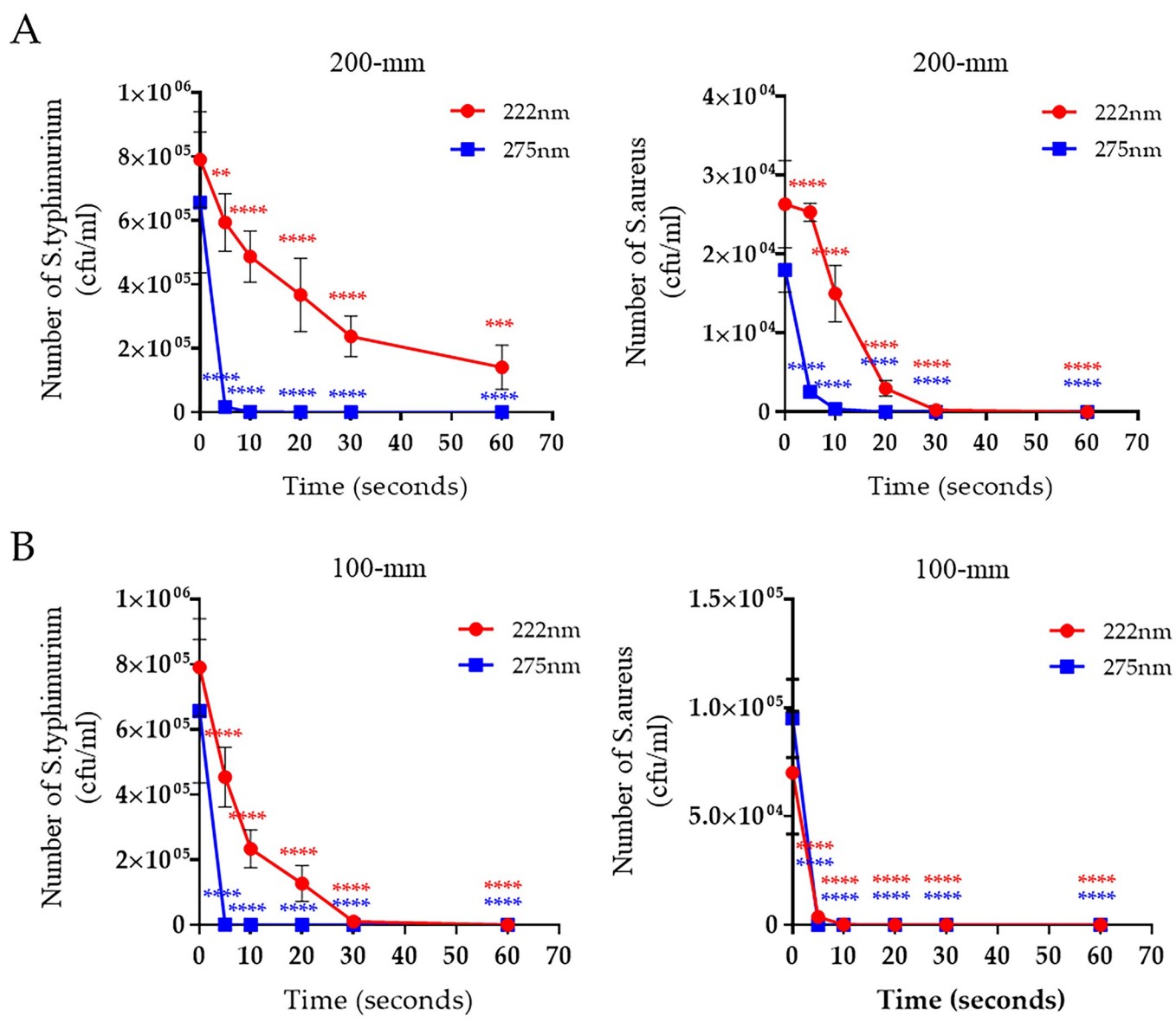

**Fig 1. Alteration in the germicidal activity of UV-C against bacteria at different irradiation distances and times.** The reduction rate of *S. typhimurium* and *S. aureus* by UV-C irradiation at distances of 200 mm (A) and 100 mm (B) during the indicated times. Representative data show the mean and standard deviation by three independent experiments. Two-way ANOVA was performed, followed by Tukey's multiple comparison tests. Statistical significance is induced by asterisks (n = 3, $**P < 0.01$, $***p < 0.001$, $****p < 0.0001$).

and SARS-CoV-2. First, virus-containing supernatant was placed on a sterile tissue culture dish and then directly irradiated with UV-C at various times and distances (liquid environment). Second, UV-C irradiation was also assessed under various conditions after absorbing the supernatant onto gauze (fabric environment). At 200-mm distance, irradiation for 10, 30, and 60 s (170, 510, and 1,020 mJ/cm$^2$, respectively) of 275-nm UV-C produced a reduction of 2.9, 4.1, and 4.1 log$_{10}$ (99.9%, 99.99%, and 99.99% reduction rate, respectively) of liquid-LPAIV. Irradiation for 10, 30, and 60 s (3.52, 10.56, and 21.12 mJ/cm$^2$, respectively) of 222-nm UV-C produced a reduction of 1.1, 1.3, and 1.6 log$_{10}$ (91.1%, 94.8%, and 97.5% reduction rate, respectively) of liquid-LPAIV (Fig 2A). The fabric-LPAIV was reduced by 1.5, 1.9, and 3.1 log$_{10}$ (97%, 98.8%, and 99.92% reduction rate, respectively) and by 0.63, 0.8, and 1 log10

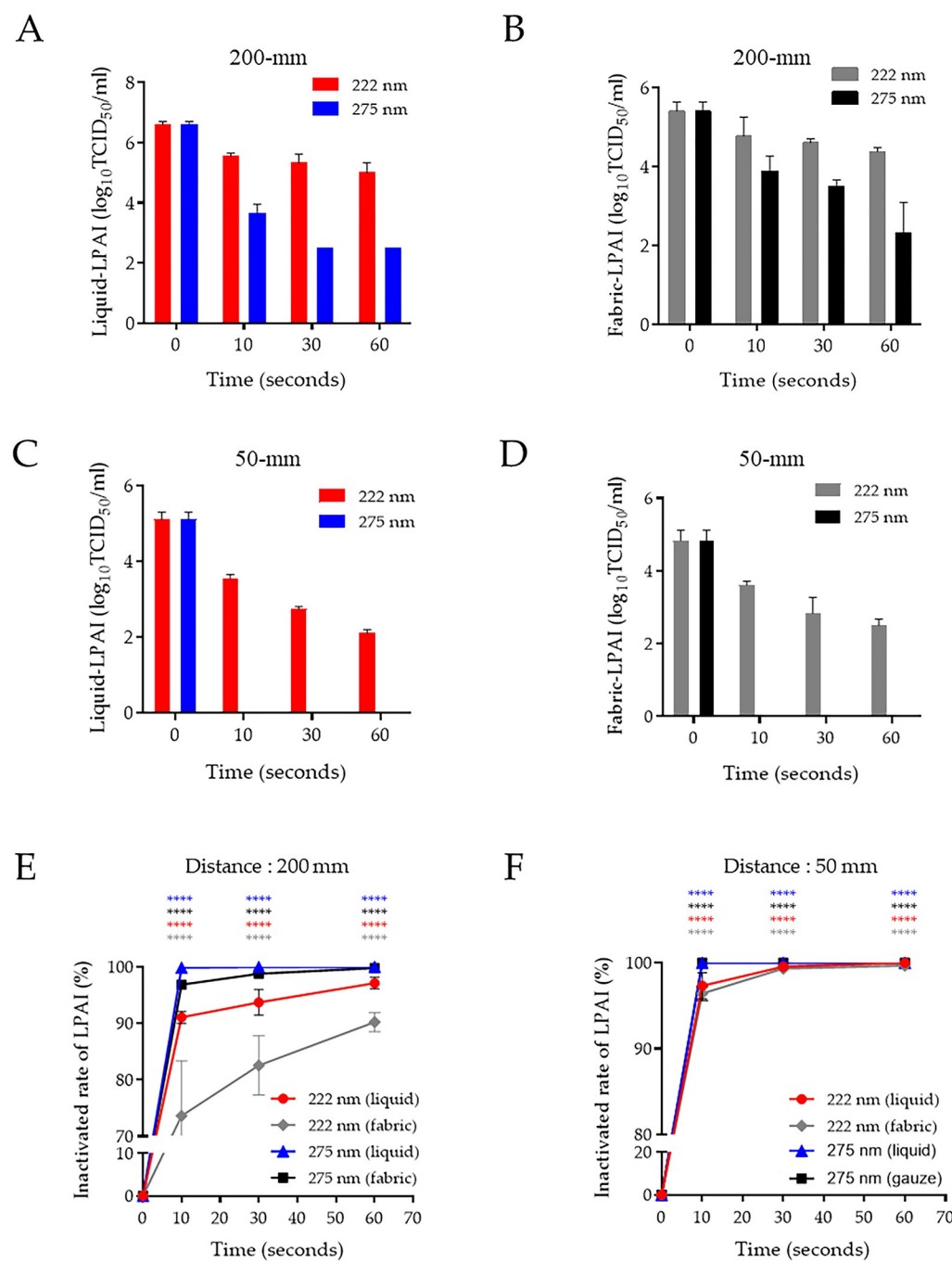

**Fig 2. Inactivation of LPAIV by UV-C irradiation in two contaminated environmental conditions following irradiation at different distances and times.** The reduction rate of liquid- and fabric-LPAIV upon UV-C irradiation at distances of 200 mm (A and B) and 50 mm (C and D) during the indicated times. Inactivation rate of liquid- and fabric-LPAIV in panels (E) and (F). Representative data show the mean and standard deviation by three independent experiments. Two-way ANOVA with Tukey's multiple comparison tests was used to determine the significance (n = 3, ****$p < 0.0001$).

(76.6%, 84.2%, 90% reduction rate, respectively) after irradiation of 275-nm and 222-nm UV-C in same distance and times (Fig 2B). The liquid- and fabric-LPAIV were inactivated by 222-nm UV-C irradiation for 10, 30, and 60 s with a 1.6, 2.3, and 3.0 $\log_{10}$ reduction,

respectively, in liquid (97.2%, 99.5%, and 99.9% reduction rate, respectively) and a 1.5, 2.3, and 2.6 $\log_{10}$ reduction, respectively, in fabric (96.8%, 99.5%, and 99.8% reduction rate, respectively) at 50-mm distance (Fig 2C and 2D). Irradiation at 50-mm with 275-nm UV-C irradiation for 10 s completely inactivated LPAIV in liquid and fabric (Fig 2B and 2C, left). These results demonstrated that 222-nm UV-C requires at least 25.1 mJ/cm$^2$ energy to show >95% virucidal activity against LPAIV. We summarized the inactivated LPAIV rate by irradiation using 222-nm and 275-nm UV-C with 200- and 50-mm distances in liquid- and fabric-contaminated conditions (Fig 2E and 2F).

## Virucidal activity of 222-nm UV-C on SARS-CoV-2-contaminated environments

We assess the virucidal activity of 222-nm UV-C against SARS-CoV-2 in liquid- and fabric-contaminated conditions. This was first conducted using the fixed 50-mm distance, which was effective in a short time against LPAIV. The SARS-CoV-2 were markedly inactivated by 222-nm UV-C irradiation at 10 and 30 s in liquid with a respective 2.1 and 4.4 $\log_{10}$ reduction (99.1% and 99.99% reduction rate, respectively) and in fabric with a respective 2.8 and 4.72 $\log_{10}$ reduction (99.9% and 99.99% reduction rate, respectively) that decreased to undetectable at 60 s in both conditions at the 50-mm distance. In contrast, irradiation for 10 s of 275-nm UV-C completely inactivated liquid- and fabric-SARS-CoV-2 (Fig 3A and 3B). We showed that SARS-CoV-2 was 99% inactivated by 222- and 275 nm UV-C irradiation for only 10 s with 50-mm distance in both liquid- and fabric-contaminated condition (Fig 3C).

Second, we investigated the infectivity of liquid-SARS-CoV-2 by measuring the expression of two viral genes: nonstructural protein 2 (NSP2) and nucleocapsid (N). When irradiated with 222-nm UV-C for 10 s at 50-mm distance, which corresponding to an energy of 25.1 mJ/cm$^2$, the viral RNA was reduced to an undetectable level in liquid-SARS-CoV-2-infected Vero E6 cells (Fig 4A). Moreover, we determined the effect of 222-nm UV-C against liquid-SARS-CoV-2 infectivity using an immunofluorescence assay. The expression of the viral spike glycoprotein (S protein) in the liquid-SARS-CoV-2 was reduced by approximately 90% during irradiation at 25.1 mJ/cm$^2$, as indicated by a reduction in green fluorescence (Fig 4B). These results indicate that SARS-CoV-2 is highly sensitive to UV-C light, and inactivation can be rapidly achieved using 222-nm UV-C.

SARS-CoV-2 variants of concern (VOCs) have been reported [21, 22] and contribute to perpetuating the pandemic. Therefore, we next established the effect of 222-nm UV-C against SARS-CoV-2 variants such as B1.617.2 (Delta) and BA.1 (Omicron). The infectivity of liquid-SARS-CoV-2 variants was measured by immunostaining of S protein after 222-nm UV-C irradiation for 10, 30, and 60 s at 50-mm distance. Irradiation for 10 s (25.1 mJ/cm$^2$) with 222-nm UV-C reduced the level of fluorescence of S protein by 58.05% (Delta) and 89.78% (Omicron) compared with that in unirradiated SARS-CoV-2 variants. Irradiation for 30 s (75.3 mJ/cm$^2$) reduced fluorescence by 98.4% (Delta) and 95.58% (Omicron); 60 s (150.6 mJ/cm$^2$) of irradiation completely inactivated both liquid-SARS-CoV-2 variants (Fig 5A). Finally, we analyzed the levels of viral proteins in Vero E6 cells infected with the irradiated liquid-SARS-CoV-2 variants to confirm virus sterilization by 222-nm UV-C. The levels of N and S protein levels of SARS-CoV-2 were significantly decreased following irradiation with 222-nm UV-C (Fig 5B). Therefore, SARS-CoV-2 variants were quickly inactivated and completely lost infectivity following irradiation with 222-nm UV-C at 75.3 mJ/cm$^2$ of 222-nm UV-C.

## Discussion

The mechanism of UV-induced sterilization is reported to induce genetic damage in DNA/RNA strands of microbial cells or viruses. The production of UV-induced cyclobutene

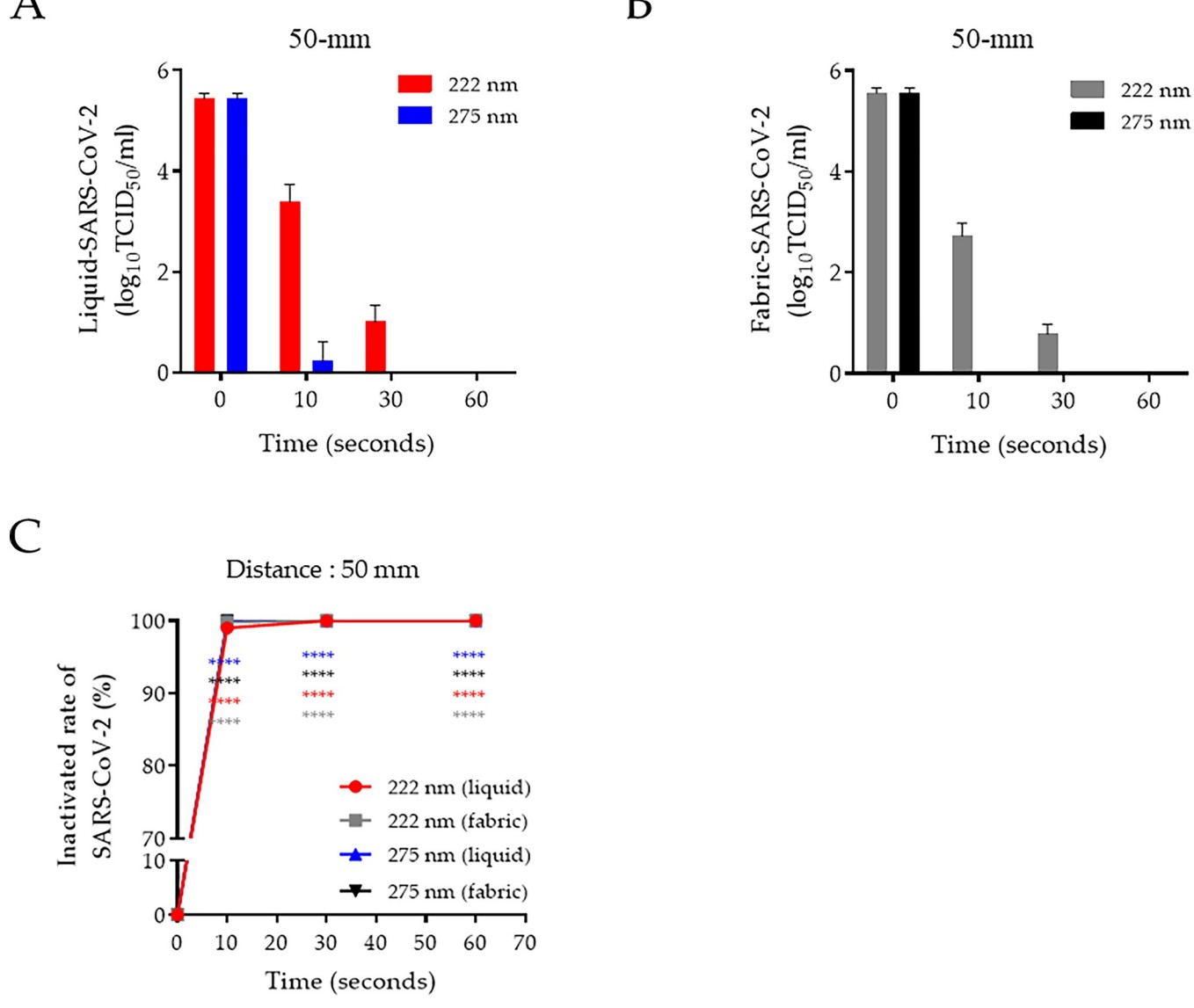

**Fig 3. Antiviral effect of UV-C against SARS-CoV-2 in two contaminated environmental conditions.** (A) Liquid- and (B) fabric-contaminated with infectious SARS-CoV-2 was irradiated with by UV-C at a distance of 50 mm during the indicated times. (C) Comparison of SARS-CoV-2 inactivation rate using 222- and 275-nm UV-C between liquid- and fabric-contaminated environments at 50-mm distance. Represent data show the mean and standard deviation by three independent experiments. Two-way ANOVA with Tukey's multiple comparison tests was used to calculate the significance (n = 3, ****$p < 0.0001$).

pyrimidine dimers causes the deformation of DNA and RNA molecules, which may induce the inactivation of cell metabolism [23]. In addition, UV irradiation generates free radicals and reactive oxygen species (ROS) with 30% penetration. The UV-induced ROS triggers several cytotoxic processes disrupting membrane integrity, DNA damage, and inactivating cellular enzymes [24, 25]. However, this UV-induced damage is affected by various UV wavelengths. In particular, UV-C light is less damaging as it cannot penetrate as deeply into tissue layers as UV-B can [26]. We considered that UV-C-induced cell damage was due dependent on the components of the cell wall and membrane. In this study, we assessed two categories of bacteria: gram-positive contains a thick peptidoglycan cell layer but no outer lipid membrane,

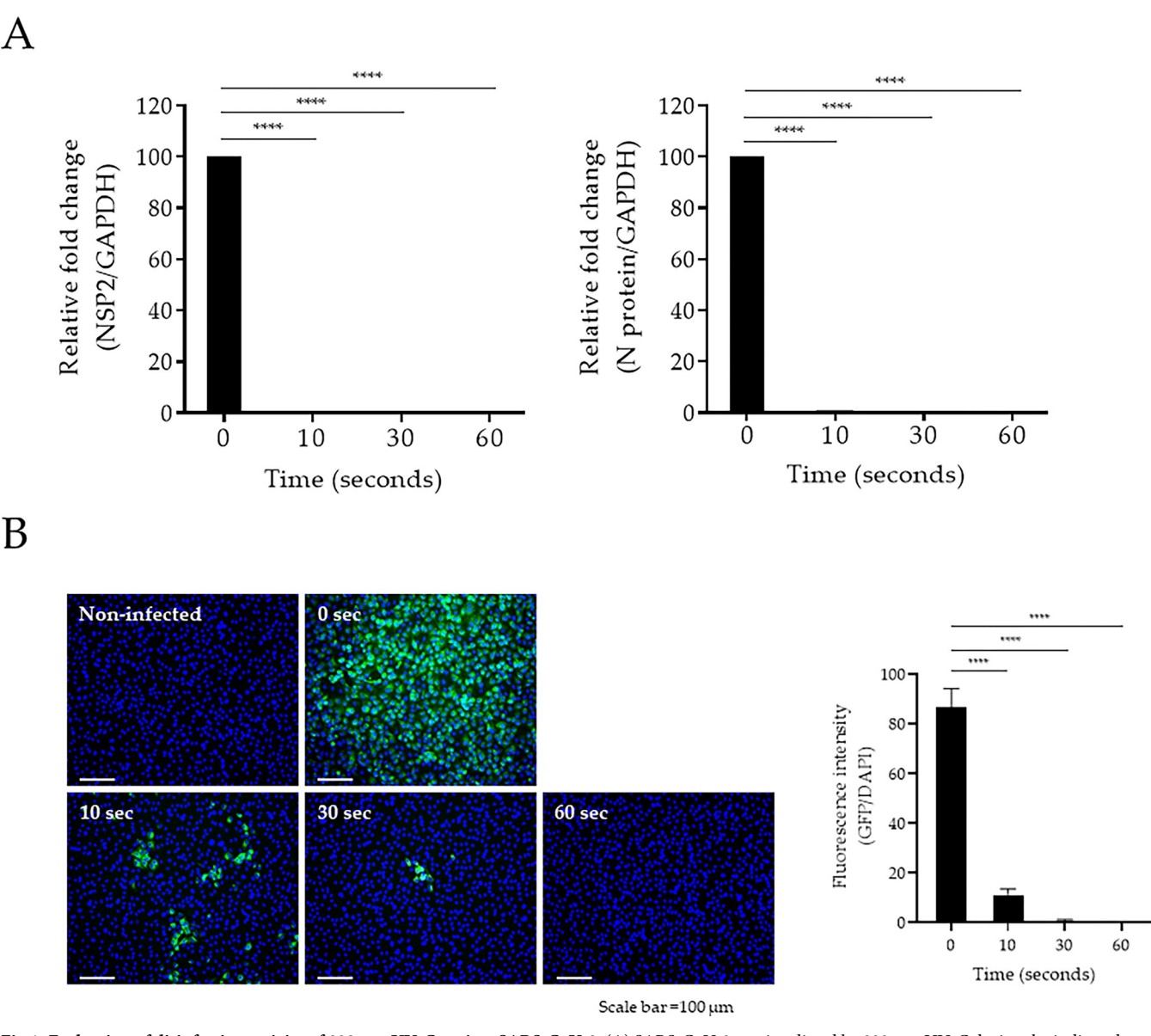

**Fig 4. Evaluation of disinfection activity of 222-nm UV-C against SARS-CoV-2.** (A) SARS-CoV-2 was irradiated by 222-nm UV-C during the indicated times at a 50-mm distance before infection into Vero E6 cells. At 24-h post-infection, total RNA was extracted from cells infected with SARS-CoV-2. Expression of the NSP2 and N genes of SARS-CoV-2 were quantified via qRT-PCR using specific primer sets. (B) Vero E6 cells infected with SARS-CoV-2 labeled with spike protein (GFP) were detected using the CELENA® S digital imaging system. The quantification of the fluorescence intensity was analyzed by GFP/DAPI using HALO image analysis software. One-way ANOVA with Tukey's multiple comparison tests was used to calculate the significance (n = 3, ****$p < 0.0001$).

whereas gram-negative contains a thin peptidoglycan layer and outer lipid membrane. Microorganisms have been reported to have different sensitivities against 222-nm UV-C [27]. Our results show that the irradiation of 275-nm UV-C within 10 s (<170 mJ/cm$^2$) was sufficient to inactivate bacteria, regardless of cell membrane compositions and distance from a lamp. In contrast, at 100-mm distance, irradiation with >27 mJ/cm$^2$ energy of 222-nm UV-C was required to inactivate *S. typhimurium*, and at least 9 mJ/cm$^2$ energy was needed to inactivate *S. aureus* (Fig 1). We considered that the 222-nm UV-C easily penetrates the peptidoglycan layer, whereas it is inhibited by a lipid membrane containing multiunit saccharides and

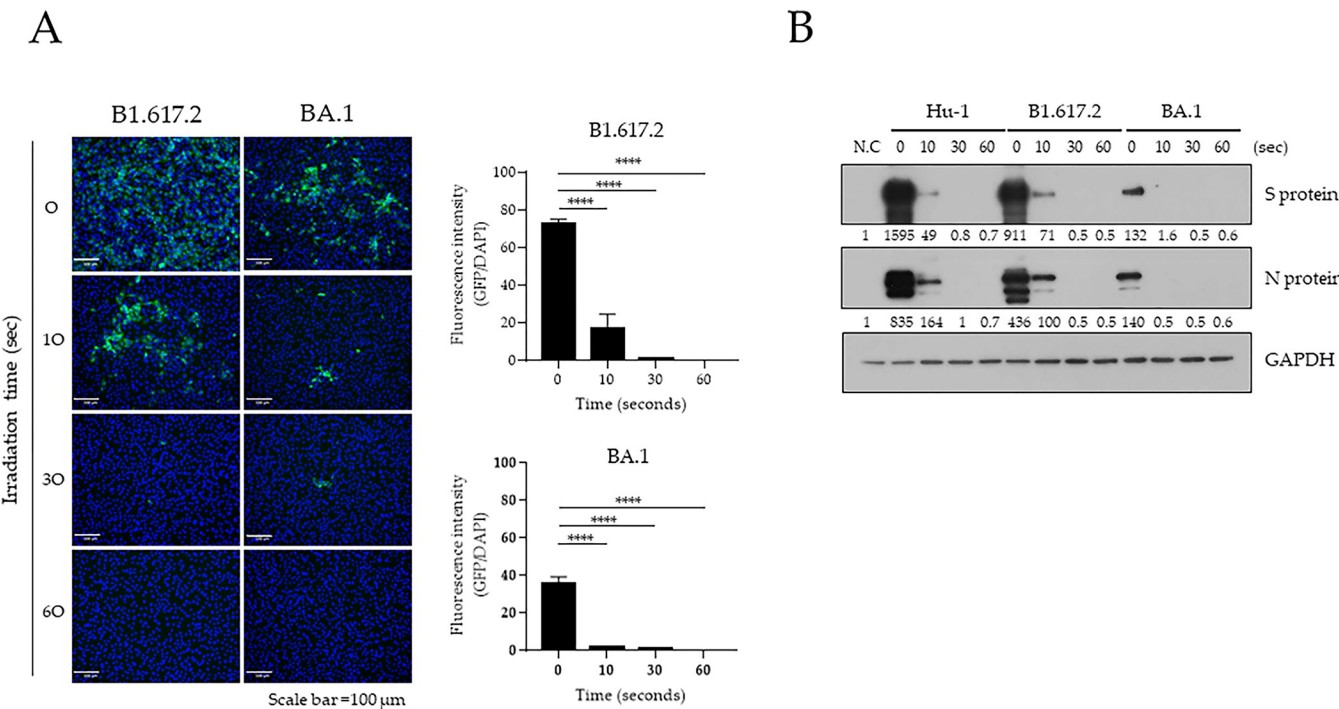

**Fig 5. Disinfecting capacity of 222-nm UV-C against SARS-CoV-2 variants.** (A) B1.617.2 (Delta) and BA.1 (Omicron) variants were irradiated with 222-nm UV-C during the indicated times at a 50-mm distance before infection into Vero E6 cells. At 24-h post-infection, cells were labeled with SARS-CoV-2 spike (S) protein and then detected using the CELENA® S digital imaging system. The quantification of the fluorescence intensity was analyzed by GFP/DAPI using HALO image analysis software. One-way ANOVA was performed, followed by Tukey's multiple comparison tests and calculation of the significance (n = 3, *$p < 0.05$, **$p < 0.01$, ***$p < 0.001$, ****$p < 0.0001$). (B) N and S proteins of SARS-CoV-2 were detected in the cell lysates using western blotting at the indicated irradiation times. Relative S and N protein expression was normalized by that of GAPDH using Image J software (version 1.53k, http://imagej.nih.gov/ij/).

proteins. The germicidal activity of 222-nm UV-C is required for suitable penetration energy based on the irradiation time and distance from a lamp against different microorganisms. The virucidal activity of 222-nm UV-C, like that of 254-nm, against various viruses, including DNA and RNA forms and enveloped and nonenveloped forms, has been reported [28–31]. However, the 222-nm UV-C mechanism for disinfecting viruses is poorly understood. The 222-nm UV-C light has minimal penetration because this light is absorbed strongly by several proteins and biomolecules compared with 254-nm UV-C [32]; however, the limited penetration of 222-nm UV-C is sufficient to traverse viruses <1 µm in size. Thus, 222-nm UV-C has virucidal activity without producing adverse side effects [33, 34]. Herein, we assessed the virucidal activity of 222-nm UV-C against potential pandemic viruses such as avian influenza and SARS-CoV-2 by testing their persistence in fabric and in liquid-contaminated environments. The avian influenza virus belongs to the type A influenza virus and is divided into high- or low-pathogenic avian influenza viruses based on the pathogenicity. Although LPAIV generally has mild symptoms, it can be transmitted to humans, causing severe illness [35]. The human-transmitted H7N9 strain as LPAIV caused a worldwide outbreak between early 2013 and 2017 [36]. UV-C has since been recognized as a valuable tool for reducing the risk of viral-contaminated environment exposure in biosecurity programs [37–39]. We tested the virucidal activity of UV-C against LPAIV in two different virus-contaminated conditions: liquid and fabric. At 200-mm distance, the 222-nm UV-C was ineffective compared with 275-nm UV-C irradiation in both contaminated conditions, even energy dosages up to 21.12 mJ/cm². Although the virucidal activity was lower in the fiber conditions than in the liquid condition, the antiviral

activity of both UV-Cs wavelengths was improved by increasing the irradiation dose by adjusting the distance (Fig 2). These differences between liquid- and fabric-LPAIV may be due to the penetration rates of UV-C based on contamination conditions. We believed that it would not be easy to target the absorbed virus in the fabric conditions because the penetration of UV-C light is disturbed by the fiber layer. Since the WHO declared the coronavirus disease (COVID-19) pandemic caused by SARS-CoV-2, there has been a continuous threat to public health [40]. SARS-CoV-2 is transmitted directly from person to person through droplets and indirectly via contaminated environments [41]. During the COVID-19 pandemic, UV disinfection, which can disinfect places or objects contaminated by SARS-CoV-2, attracted attention and many products have been developed [42]. The comprehensive use of UV-C devices is expected to limit SARS-CoV-2 spread in hospitals and public healthcare facilities. Many experimental studies strongly support that UV-C irradiation rapidly inactivates infectious SARS-CoV-2 [43, 44]. Our results confirmed that 222-nm UV-C at an energy of 25.1 mJ/cm$^2$ efficiently inactivated SARS-CoV-2 (including recent VOC) in liquid and fabric environments. The virucidal activity of UV-C affects the viral genome and thus can occur regardless of which SARS-CoV-2 variant is present. Typically, the reproduction of RNA viruses is prone to error because of the lack of proofreading machinery. The error rates may be close to those that kill the virus. For this reason, genetic damage to a single-stranded viral genome by UV-C irradiation is thought to be more lethal. Therefore, 222-nm UV-C can be applied in public places such as airports, hospitals and medical facilities and is effective and safe. In conclusion, we investigated the disinfection efficacy of 222nm UV-C against both gram-negative and -positive bacteria and two potential pandemic viruses through two different methods. Our results suggest that 222-nm UV-C is a useful disinfection tool against various pathogens and should be further investigated in studies on safety and effectiveness for practical applications.

## Supporting information

**S1 Table. Survival rate of cell numbers by irradiated UV-C with several optical dosages.** (PPTX)

**S1 Raw image.** (PDF)

## Acknowledgments

Information on the UV-C from the Biodech company used in this study is provided in the supporting information. The authors thank the staff from the Biosafety-Level 3 facility (Korea Zoonosis Research Institute, Jeonbuk National University). All experiment raw data included in this study could be available upon request by containing the corresponding author Sang-Min Kang via email: sangminkang@jbnu.ac.kr

## Author Contributions

**Conceptualization:** Sang-Min Kang, Dongseob Tark.

**Data curation:** Eun-Gyeong Lee, Kyunga Ryu, Sang-Min Kang.

**Formal analysis:** Byeong-Min Song, Gun-Hee Lee, Hee-Jeong Han, Ju-Hee Yang.

**Funding acquisition:** Gun-Hee Lee, Sang-Min Kang, Dongseob Tark.

**Investigation:** Kyunga Ryu, Jinwoo Kim.

**Methodology:** Byeong-Min Song, Gun-Hee Lee, Sang-Min Kang.

**Project administration:** Hyunji Gu, Ha-Kyeong Park.

**Supervision:** Sang-Min Kang, Dongseob Tark.

**Validation:** Gun-Hee Lee, Dongseob Tark.

**Writing – original draft:** Byeong-Min Song, Gun-Hee Lee.

**Writing – review & editing:** Sang-Min Kang, Dongseob Tark.

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
