## [Decision Letter · Decision Letter 0]

3 Oct 2023

PONE-D-23-28082Ultraviolet-C light at 222 nm has a high disinfecting spectrum in environments contaminated by infectious pathogens, including SARS-CoV-2PLOS ONE

Dear Dr. Kang,

Thank you for submitting your manuscript to PLOS ONE. After careful consideration, we feel that it has merit but does not fully meet PLOS ONE’s publication criteria as it currently stands. Therefore, we invite you to submit a revised version of the manuscript that addresses the points raised during the review process.

We look forward to receiving your revised manuscript.

Kind regards,

Yong Sam Jung, Ph.D.

Academic Editor

PLOS ONE

Journal Requirements:

   "This research was supported by the Basic Science Research Program through the National Research Foundation of Korea (NRF) funded by the Ministry of Education (2021R1C1C2005307 and 2017R1A6A1A03015876) and supported by the grant of the Korea Health Industry Development Institute (KHIDI, HI22C1637). 

S.K. received research funding, HI22C1637, from KHIDI,  G.L., and D.T. received 2021R1C1C2005307 and 2017R1A6A1A03015876 from NRF. "

7. PLOS ONE now requires that authors provide the original uncropped and unadjusted images underlying all blot or gel results reported in a submission’s figures or Supporting Information files. This policy and the journal’s other requirements for blot/gel reporting and figure preparation are described in detail at https://journals.plos.org/plosone/s/figures#loc-blot-and-gel-reporting-requirements and https://journals.plos.org/plosone/s/figures#loc-preparing-figures-from-image-files. When you submit your revised manuscript, please ensure that your figures adhere fully to these guidelines and provide the original underlying images for all blot or gel data reported in your submission. See the following link for instructions on providing the original image data: https://journals.plos.org/plosone/s/figures#loc-original-images-for-blots-and-gels. 

8. Please include your tables as part of your main manuscript and remove the individual files. Please note that supplementary tables (should remain/ be uploaded) as separate "supporting information" files

9. We notice that your supplementary figures are uploaded with the file type 'Figure'. Please amend the file type to 'Supporting Information'. Please ensure that each Supporting Information file has a legend listed in the manuscript after the references list.

Additional Editor Comments:

When revising your manuscript, please consider all issues mentioned in the reviewers' comments carefully: please outline every change made in response to their comments and provide suitable rebuttals for any comments not addressed. Please note that your revised submission may need to be re-reviewed. 

Reviewers' comments:

Reviewer's Responses to Questions

**Comments to the Author**

1. Is the manuscript technically sound, and do the data support the conclusions?

Reviewer #1: Yes

Reviewer #2: Yes

Reviewer #3: Partly

2. Has the statistical analysis been performed appropriately and rigorously? 

Reviewer #1: Yes

Reviewer #2: Yes

Reviewer #3: Yes

3. Have the authors made all data underlying the findings in their manuscript fully available?

Reviewer #1: Yes

Reviewer #2: Yes

Reviewer #3: Yes

4. Is the manuscript presented in an intelligible fashion and written in standard English?

Reviewer #1: Yes

Reviewer #2: Yes

Reviewer #3: Yes

5. Review Comments to the Author

Reviewer #1: In this manuscript, authors investigated the effect of UVC at 222nm for disinfection of pathogens, particularly SARS-Cov-2, comparing with 275nm UVC. Authors clearly showed that the 222nm UVC exerts the the germicidal and virucidal activity against SARS-cov-2 and couple of bacteria in various contaminating conditions (liquid, fabric) and can effectively inactivates SARS-cov-2 variants such as Delta and Omicron. Considering the significance of pandemic situation of infectious viruses such as Flu and Corona and versatility/safety of 222nm UVC, these data support the powerful preventive strategy against various pathogen. Nevertheless, the manuscript would be further strengthened if the authors can address the following minor concern.

1. In introduction section, author need to clearly clarify the aim of this study why they performed the experiment in liquid and fabric condition, not aerosol and compared the effect in various exposure time, etc. And it also need to be discussed.

Reviewer #2: This manuscript describes the use of Ultraviolet-C light at 222 nm has a high disinfecting spectrum in environments contaminated by infectious pathogens, including SARS-CoV-2 by Song. They examined the effect of 222 nm UV-C irradiation on contaminating bacteria and viruses. A colony forming unit assay was used to determine the number of viable cells, and a virus titration was used as the assay of evaluation. They also found that the irradiation effectively inactivated Delta and Omicron, variants of SARS-CoV-2. These results provide valuable information on the bactericidal efficiency of 222nm UV-C in environments contaminated with bacteria and viruses. The experimental methods, discussion, and argument of the paper are very well done.

Reviewer #3: Authors demonstrated the disinfection efficiency of 222-nm UV-C in bacterial and virus-contaminated environments. Overall, the manuscript was conducted properly and well-written. And the results may provide the valuable information on disinfection. However, the finding lacks the novelty, because the disinfection efficiency of UV-C is already known. The authors did not describe the reason why that they selected 222-nm UV-C in their study. Generally, the disinfection efficiency of 222-nm UV-C is lower than that of 275-nm UV-C. There is no information supporting the safety and advantage of 222-nm UV-C. In virucidal study, the authors only tested enveloped RNA viruses. The authors may need to test other types of viruses such as non-enveloped or DNA viruses.

6. PLOS authors have the option to publish the peer review history of their article (what does this mean?). If published, this will include your full peer review and any attached files.

Reviewer #1: No

Reviewer #2: No

Reviewer #3: No

---

## [Author Response · Author response to Decision Letter 0]

20 Oct 2023

We considered the reviewer's comments favorably and did our best to respond appropriately. I am sending you a "Response to reviewers" file to the editor, so please let me know if there is anything I need to revise.

---

## [Editor Report · Decision Letter 1]

31 Oct 2023

Ultraviolet-C light at 222 nm has a high disinfecting spectrum in environments contaminated by infectious pathogens, including SARS-CoV-2

PONE-D-23-28082R1

Dear Dr. Kang,

We’re pleased to inform you that your manuscript has been judged scientifically suitable for publication and will be formally accepted for publication once it meets all outstanding technical requirements.

Kind regards,

Yong Sam Jung, Ph.D.

Academic Editor

PLOS ONE

---

## [Editor Report · Acceptance letter]

15 Nov 2023

PONE-D-23-28082R1 

Ultraviolet-C light at 222 nm has a high disinfecting spectrum in environments contaminated by infectious pathogens, including SARS-CoV-2 

Dear Dr. Kang:

I'm pleased to inform you that your manuscript has been deemed suitable for publication in PLOS ONE. Congratulations! Your manuscript is now with our production department. 

Kind regards, 

on behalf of

Dr. Yong Sam Jung 

Academic Editor

PLOS ONE